# The Law of Parsimony and the Negative Charge of the Bubbles

**Stoyan I. Karakashev *** and **Nikolay A. Grozev**

Department of Physical Chemistry, Sofia University, 1 James Bourchier Blvd, 1164 Sofia, Bulgaria; fhng@chem.uni-sofia.bg
* Correspondence: fhsk@chem.uni-sofia.bg; Tel.: +359-28161283

**Abstract:** Why the bubbles are negatively charged? This is almost 100 years old question, which many scientists have striven and still are striving to answer using the latest developments of the MD simulations and various physical analytical methods. We scrutinize with this paper the basic literature on this topic and conduct our own analysis. Following the philosophical law of parsimony: "Entities should not be multiplied without necessity", we assume that the simplest explanation is the right one. It is well known that the negative change of the Gibbs free energy is a solid criterion for spontaneous process. Hence, we calculated the energies of adsorption of $OH^-$, $H_3O^+$ and $HCO_3^-$ ions on the air/water interface using the latest theoretical developments on the dispersion interaction of inorganic ions with the air/water interface. Thus, we established that the adsorption of $OH^-$ and $HCO_3^-$ ions is energetically favorable, while the adsorption of $H_3O^+$ is energetically unfavorable. Moreover, we calculated the change of the entropy of these ions upon their transfer from the bulk to the air/water interface. Using the well-known formula $\Delta G = \Delta H - T\Delta S$, we established that the adsorption of $OH^-$ and $HCO_3^-$ ions on the air/water interface decreases their Gibbs free energy. On the contrary, the adsorption of $H_3O^+$ ions on the air/water interface increases their Gibbs free energy. Thus, we established that both $OH^-$ and $HCO_3^-$ ions adsorb on the air/water interface, while the $H_3O^+$ ions are repelled by the latter. Therefore, electrical double layer (EDL) is formed at the surface of the bubble–negatively charged adsorption layer of $OH^-$ and $HCO_3^-$ ions and positively charged diffuse layer of $H_3O^+$ ions.

**Keywords:** ion-specific effects; DI water; adsorption of ions; bubbles

---

## 1. Introduction

The properties of the air/water interface have been a puzzle to solve for many years. Heydweiller [1] was the first to report in 1910 that the addition of inorganic salt increases the surface tension of water with a certain constant increment specific for each salt. Wagner in 1924 [2] and Onsager and Samaras in 1936 [3] explained these experimental observations for the first time with the existence of image repulsion forces between the ions of the salt and air/water interface, which cause their negative adsorption and consequently higher surface tension. Schwenker [4] conducted additional measurements on the surface tension of salt solutions in 1931 and established deviations from the measurements of Heydweiller for the case of diluted salt solutions. To better study this puzzle, Jones and Ray [5,6] conducted a unique experiment in 1937 and 1941 with great precision on the surface tension of diluted salt solutions. For this purpose, they built a special tensiometer, called a capillarimeter. They established that the inorganic salts decrease the surface tension of water until reaching a minimum (≈0.0002% drop) within the concentration range of 0.001–0.002 mol/L, beyond which the surface tension increases with steady increment. Hence, the surface tension increment changes specifically for each salt substantially in value and sign in the concentration range of 0.0005–0.002 mol/L [1,4]. Instead of solving the

puzzle, they declared a scientific problem, which remained unsolved until the last two decades of XXI century, when the molecular dynamic simulations (MDS) and the computers underwent significant development.

Another puzzle of the air/water interface is its negative charge. Lenard was the first to establish in 1915 [7] that the water droplets are negatively charged, and studied the possible reason for such a charge [8]. In those works, Lenard arrived at the conclusion that the very surface layer of water is electrically negatively charged, and there is a positively charged layer at a lower depth. The possible reason for such a distribution of the charges at the very air/water interface remained unclear until now. Nevertheless, intensive scientific efforts were put forward to unveil this mystery. Exerowa was the first to study in 1969 the effect of the pH value on the surface potential of the air/water interface [9]. She confirmed the negative charge of the bubbles and established that their isoelectric point (IEP) is at pH ≈ 4. Hence, she hypothesized that the $OH^-$ ions adsorb on the surface of the bubbles, thus charging them negatively. Unfortunately, she and her followers were not able to explain what makes the $OH^-$ ions adsorb on the air/water interface. Since that time, many authors [10–21] studied the zeta potential of the air bubbles in water under various conditions. All of them confirmed that the negative zeta potential of the bubbles is sensitive to the pH and ionic strength values. The nonionic surfactants practically do not affect the zeta potential of the bubbles, while the ionic surfactants do. They also confirmed that the isoelectric point (IEP) is in the range of pH = 3–4. Greay-Weale and Beattie [22,23] and Bai and Herzfeld [24] suggested that the autolysis of the surface water is different from this one of the bulk water, thus resulting in positive adsorption of both $H_3O^+$ and $OH^-$ ions on the air/water interface, but the surface concentration of $OH^-$ ions is significantly larger than that of the $H_3O^+$ ions at the interface boundary. Kallay et al. [25] suggested thermodynamic model of the air/water interface based on this assumption. Unfortunately, this assumption remained just nice hypothesis. Ruckenstein and Manciu [26,27] suggested a model pairing surface tension and zeta potential taking into account all the possible ions into the water with variable pH, but the model contained adjustable parameters obtained from the experimental zeta potential of the bubbles. Resulting in negative zeta potential of the bubbles, unfortunately substantially undervalued. Duignan et al. [28] reported that traces of anionic surface-active contaminant charge the bubbles negatively and the addition of salt up to 1–2 mM reduces the surface tension due to additional adsorption of the surface-active contaminant. This imposes the question whether the purest possible deionized (DI) water contains traces of surface-active substances, and if yes, why all of them are negatively charged? To shed a light on this mystery, the molecular dynamic simulations (MDS) paired with powerful computers were curbed in the last two decades. The majority of molecular dynamic simulations (MDS) of the air/water interface [29–33] have shown that the $OH^-$ ions should not be adsorbed at the air/water interface, but $H_3O^+$ ions should, thus charging the bubbles positively. Few authors argued that their MDS analysis [34,35] shows the opposite. Poli et al. [36] conducted quantum-chemical calculations in 2020 showing that the negative charge of the bubbles originates from a charge transfer from the bulk water molecules towards the surface water molecules by means of the topologically defected hydrogen bond network. This solution looks nice, but it does not explain the pH dependence of the zeta potential of the bubbles. Therefore, one can see the enormous scientific effort to solve this puzzle resulting in various complex explanations.

Tabor et al. [37] conducted a unique experiment on the stability of bubbles from carbon dioxide in contact and showed that pairs of such bubbles in contact live significantly longer than pairs of bubbles from other gases (air, argon, nitrogen). They explained this observation with the adsorption of $HCO_3^-$ ions on the surfaces of the bubbles, originating from the carbon dioxide, thus charging them and stabilizing them electrostatically. Yan et al. [38] conducted another unique experiment, thus showing that $HCO_3^-$ ions prevent the aggregation of the hydrophobic Poly(methyl methacrylate) particles and oil droplets. These few experiments show unambiguously that the $HCO_3^-$ ions adsorb on hydrophobic surfaces (air/water, oil/water, hydrophobic solid/water). The intrinsic deionized (DI) water contains

basically $OH^-$, $HCO_3^-$, and $H_3O^+$ ions, and the concentrations of all of them are pH sensitive. Do they adsorb on the surfaces of the bubbles?

This long-standing mystery motivated us to conduct our own analysis. Indeed, the decrease of the surface tension at small concentrations of inorganic salts [5,6] can be conjugated with the positive adsorption of ions. Thus, we arrived at the conclusion that some ions are adsorbed at the air/water interface, thus charging the latter, while other ions remain at a lower depth, forming a diffusive layer. Both of them form electrical double layer (EDL), as is known from textbooks [39–41]. To estimate the adsorption propensity of $OH^-$, $H_3O^+$ and $HCO_3^-$ ions at the air/water interface, we exploited the theory of Ivanov [42,43] of the specific adsorption energy of inorganic ions at the air/water interface. Thus, we calculated the adsorption energies of the $OH^-$, $HCO_3^-$, and $H_3O^+$ ions on the air/water interfaces along with their entropy change of transfer from the 3D bulk to the 2D air/water interface. Therefore, we calculated the change of the Gibbs free energy of adsorption of these three ions on the air/water interface and identified if their adsorption is spontaneous or not. In this way, we applied the ancient law of parsimony stating that the simplest solution of given dilemma is the right one among many other complex solutions.

## 2. Theoretical Review

### 2.1. Speciation Analysis of the Deioniozed (DI) Water

The intrinsic pH value of the deionized (DI) water is in the range of $5.6 \leq pH \leq 6$ (in most cases $pH \approx 5.8$) [44]. This is due to the dissolved carbon dioxide ($CO_2$) from the air. Therefore, the DI water is a solution of carbonic acid. The following equilibria are valid for such a solution [45–47]:

$$(CO_2)_{aq} + H_2O \overset{K_r}{\rightleftarrows} H_2CO_3, K_r = 1.7 \times 10^{-3} \tag{1}$$

$$H_2CO_3 + H_2O \overset{K_{a1}}{\rightleftarrows} H_3O^+ + HCO_3^-, K_{a1} = 2.5 \times 10^{-4} \tag{2}$$

$$HCO_3^- + H_2O \overset{K_{a2}}{\rightleftarrows} H_3O^+ + CO_3^{2-}, K_{a2} = 4.69 \times 10^{-11} \tag{3}$$

$$2H_2O \overset{K_w}{\rightleftarrows} OH^- + H_3O^+, K_w = 10^{-14} \tag{4}$$

The above processes are related to the following analytical relations:

$$K_{a1} = \frac{C_{HCO_3^-} C_{H_3O^+}}{C_{H_2CO_3}} \tag{5}$$

$$K_{a2} = \frac{C_{CO_3^{2-}} C_{H_3O^+}}{C_{HCO_3^-}} \tag{6}$$

$$K_w = C_{H_3O^+} C_{OH^-} \tag{7}$$

$$C_{H_3O^+} = C_{HCO_3^-} + C_{OH^-} + 2C_{CO_3^{2-}} \tag{8}$$

$$C_0^{H_2CO_3} = C_{H_2CO_3} + C_{HCO_3^-} + C_{CO_3^{2-}} \tag{9}$$

where $K_{\alpha 1} = 2.5 \times 10^{-4}$ mol/L and $K_{\alpha 2} = 4.69 \times 10^{-11}$ mol/L are the dissociation constants of the carbonic acid, while $K_w = 10^{-14}$ is the ionic constant of the water at temperature 20 °C. We calculated the concentrations of $OH^-$, $HCO_3^-$, and $H_3O^+$ ions by means of Equations (5)–(9) and considering pH = 5.8. All the data needed for these calculations were taken from Reference [45]. Thus obtained, the concentrations were used for calculating the entropies of bulk ions.

### 2.2. Energy of Adsorption of Ions on the Air/Water Interface of Diluted Aqueous Salt Solutions

We shall present the theory calculating the specific adsorption energies of the ions on the air/water interface [42,43] as developed by Ivanov in his effort to study the specific effect of the ions on the adsorption layer of ionic surfactants [42]. The very development of this theory does not consider the presence of surfactant. Therefore, it is applicable for neat (surfactant free) air/water interface. It is valid for diluted salt solutions up to 2 mM concentration similar to Poisson–Boltzmann equation in its classical form. It is based on the difference of the energy of two basic states (see Figure 1): (i) ion located in the bulk of water; (ii) ion located on the air/water interface. The state (i) accounts for the dispersion energy of interaction $u_{iw}^b$ of a bulk ion (signified by subscript "$i$") with the whole bulk of water concurrently with the dispersion energy of interaction $u_{ww}^s$ of ensemble of $N_w$ surface water molecules, suitable in a volume equal to the volume of the bulk ion, with the bulk of water underneath the interphase boundary. These $N_w$ surface water molecules are supposed to be displaced by the ion upon its adsorption on the air/water interface. The state (ii) accounts for the dispersion energy of interaction $u_{iw}^s$ of surface ion located on the air/water interface with the bulk of water underneath concurrently with the dispersion energy of interaction $u_{ww}^b$ of the ensemble $N_w$ bulk water molecules with the whole bulk of water (see Figure 1). The specific energy of adsorption of ion on the air/water interface is defined by the difference of the energies of states (ii) and (i):

$$u_{i0} = (u_{iw}^s + u_{ww}^b) - (u_{iw}^b + u_{ww}^s) = \Delta u_i - \Delta u_w \tag{10}$$

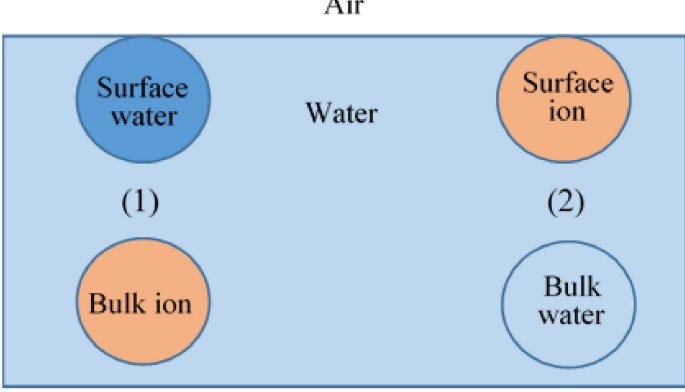

**Figure 1.** Two basic energetical states: (**1**) bulk ion and $N_w$ surface water molecules; (**2**) surface ion and $N_w$ bulk water molecules.

One can see in Equation (10) that the specific energy of adsorption $u_{i0}$ of the ion can be presented as the difference between $\Delta u_i$ and $\Delta u_w$ given by $\left(u_{iw}^s - u_{iw}^b\right) - \left(u_{ww}^b - u_{ww}^s\right)$.

The development of the theory in its original form is situated in the Appendix A of this paper. We shall present hereafter the relations for calculation of the specific adsorption energy of the ions on the air/water interface. The theory accounts for the deformability of the hydration shells of the ions at the air/water interface. Ions with tough hydration shell are usually called kosmotropic (Li$^+$, Na$^+$, OH$^-$, H$_3$O$^+$, F$^-$, etc.), and their specific adsorption energy on the air/water interface can be presented by the following equation:

$$u_{i0} = \frac{\pi}{6} \frac{\rho_w}{R_{ih}^3} \left(L_{iw} - L_{ww}\right) \tag{11}$$

Ions with deformable hydration shell are usually called chaotropic ($K^+$, $NH_4{}^+$, $Cl^-$, $Br^-$, etc.), and their specific adsorption energy on the air/water interface can be presented by the following equation:

$$u_{i0} = \left(1 - \frac{3}{4}\frac{R_{ib}}{R_{ih}}\right)\frac{2\pi}{3}\frac{\rho_w}{R_{ih}^3}(L_{iw} - L_{ww})$$

(12)

where $\rho_w$ is density of the water, $R_{ih}$ is the hydration radius of ion "$i$", $L_{iw}$ and $L_{ww}$ are London constants of ion–water and water–water dispersion interaction given by the equations:

$$L_{iw} = \frac{3\alpha_i\alpha_w}{2}\frac{I_iI_w}{I_i + I_w}, \ L_{ww} = \frac{3}{4}N_w\alpha_w^2I_w$$

(13)

where $\alpha_i$ and $\alpha_w$ are the polarizabilities of the ion "$i$" and the water molecule, while $I_i$ and $I_w$ are their ionization potentials, $N_w$ is a number of the surface water molecules, which are displaced due to adsorption of one ion on the air/water interface.

*2.3. Entropy Change of Adsorption of Ions on the Air/Water Interface*

We will calculate the entropies of the ions in the bulk and on the surface. Thus, we will establish what is the change of the entropy if they adsorb on the surface. Hence, once we have the enthalpy and the entropy of adsorption calculated, we will calculate the change of the Gibbs free energy.

The Boltzmann formula for the entropy is well known [48]:

$$S = k_B \ln W$$

(14)

where $k_B$ is the Boltzmann constant ($k_B = 1.380 \times 10^{-23}$ J/K in SI system), and $W$ is the thermodynamic probability of a given system to reside in a given macro-state. We will observe the ions of the salt as a system, which is located either in the bulk or on the surface. They are in a state of continuous motion despite being either in the bulk with diffusion coefficient $D$ or on the air/water interface with surface diffusion coefficient $D_S = 1.5D$ [49]. The detailed development of the theory for calculation of the entropy for our particular case is presented in the Appendix A. The entropy of the ions in the bulk ($S_{(3D)}$ for the 3D case) and on the air/water interface ($S_{(2D)}$ for the 2D case) is given by the following formulas:

$$S_{(3D)} = \frac{3}{2}\nu k_B\left[\ln\nu + \ln\left(\frac{m}{2k_BT\pi}\right) - \frac{m\overline{c_{(3D)}^2}}{3k_BT}\right]$$

(15)

$$S_{(2D)} = \frac{3}{2}\Gamma k_B\left[\ln\Gamma + \ln\left(\frac{m}{2k_BT\pi}\right) - \frac{m\overline{c_{(2D)}^2}}{3k_BT}\right]$$

(16)

where $\nu$ and $\Gamma$ are the bulk concentration and the adsorption of the ions on the air/water interface, $m$ is the mass of one ion, $\overline{c_{(3D)}^2}$ and $\overline{c_{(2D)}^2}$ are the average squared speed of one ion in the bulk and on the air/water interface, $k_B$ is the Boltzmann constant, and $T$ is absolute temperature.

Finally, we calculate the change of the entropy during the adsorption of the ions on the air/water interface by the expression:

$$\Delta S = S_{(2D)} - S_{(3D)}$$

(17)

The transfer of $N_w$ water molecules from the air/water interface into the bulk is related with an increase of the entropy, which formally should be accounted for in our observation. We assume this change of the entropy negligible, because the water molecules do not move neither on the air/water interface nor into the bulk due to their participation in the hydrogen bond network. The water molecules as moieties of the hydrogen bond network only vibrate along and across this network. In contrast to

them the ions diffuse into the whole bulk and move on the air/water interface as well. For this reason, we account only for the change of the ion's entropy during their adsorption.

The Gibbs free energy of the adsorption of the ions on the air/water interface will be calculated via the well-known formula [48]:

$$\Delta G = \Delta H - T \Delta S \tag{18}$$

## 3. Results and Discussion

### 3.1. Speciation Analysis of the Dioniozed (DI) Water

Our speciation analysis is based on the proceeses described in Section 2.1.

Table 1 presents the basic species and their concentrations in the deionized water. We should mention that the DI water contains dissolved oxygen ($O_2$) and nitrogen ($N_2$) as well but their presence is not important for the present analysis. The basic three ions, in which we are interested, are $OH^-$, $HCO_3^-$, and $H_3O^+$. One can see that the concentration of $HCO_3^-$ ions is about 250 times larger than the concentration of $OH^-$ ions. The concentration of $CO_3^{2-}$ ions is almost 34,000 times smaller than the concentration of $HCO_3^-$ ions. In addition, the concentrations of $HCO_3^-$ and $H_3O^+$ ions are close to each other. This means that most probably the basic effects on the surface of the bubbles will originate from the $HCO_3^-$ and the $1.585 \times 10^{-6}$. Nevertheless, we do not underestimate the possible effect of the $OH^-$ ions.

**Table 1.** Basic species and their concentrations in the deionized (DI) water.

| Specie | $OH^-$ | $HCO_3^-$ | $CO_3^{2-}$ | $H_3O^+$ | $H_2CO_3$ | $CO_2$ |
|---|---|---|---|---|---|---|
| Concentration, mol/L | $6.310 \times 10^{-9}$ | $1.579 \times 10^{-6}$ | $4.670 \times 10^{-11}$ | $1.585 \times 10^{-6}$ | $1.000 \times 10^{-8}$ | $1.360 \times 10^{-5}$ |

### 3.2. Energy of Adsorption of $OH^-$, $HCO_3^-$, and $H_3O^+$ Ions on the Air/Water Interface

We used Equations (11) and (12) to calculate the adsorption energies of the $OH^-$, $HCO_3^-$ and $H_3O^+$ ions on the air/water interface. The ions with bare radius below 1.35 Å are kosmotropic [43]. Exclusion makes the acetic ion, whose shape is more complex [43]. The kosmotropic ions undergo stronger image repulsion by the air/water interface than the chaotropic ions. The reason is their stronger electrostatic field is in close proximity, making their hydration shells tougher than the ones of the chaotropic ions [50]. Therefore, $OH^-$ and $H_3O^+$ are kosmotropic, while $HCO_3^-$ is chaotropic ion.

The adsorption of a given ion on the air/water interface causes two opposite energetic effects:(i) gain of energy $E_1$ (see Figure 2) via the adsorption of the ions by overcoming the image repulsion force; this process is endothermic; (ii) loss of energy $E_2$ (see Figure 2) due to displacement of an ensembles of surface water molecules by the ions into the bulk of water; this process is exothermic. The adsorption of the ions on the air/water interface always displaces surfaces water molecules into the bulk (see Figure 2). The total change of energy $E_{ads}$ from these two processes determines if the adsorption is favorable energetically or not. Such an approach was missing in the literature. The authors traditionally studied the dispersion interaction of the different ions with the air/water interface, thus establishing their segregation at close proximity of the phase boundary [2,3,51–60]. As far as the image force between the ion and air/water interface is always repulsive, they arrived at the conclusion that all the ions are repelled by the air/water interface but at different degrees—some ions are more repelled than other ions, thus causing their segregation at the very surface. These results were well paired with the experimental increase of the surface tension at large salt concentration. In addition, they were confirmed by the more precise molecular dynamic (MD) simulations of salt solutions close to the air/water interface [61–64]. All these literature studies cannot explain the Ray–Jones effect; neither they can explain the negative origin of the air/water interface. The reason is that they studied salt solutions with high concentration ($c > 0.5$ mol/L). Our approach is traditional as well but more general. We are not interested in the segregation of the different ions close to the interphase boundary.

The calculation of their adsorption energy $E_{\text{ads}}$ despite being either positive or negative is more practical and definite.

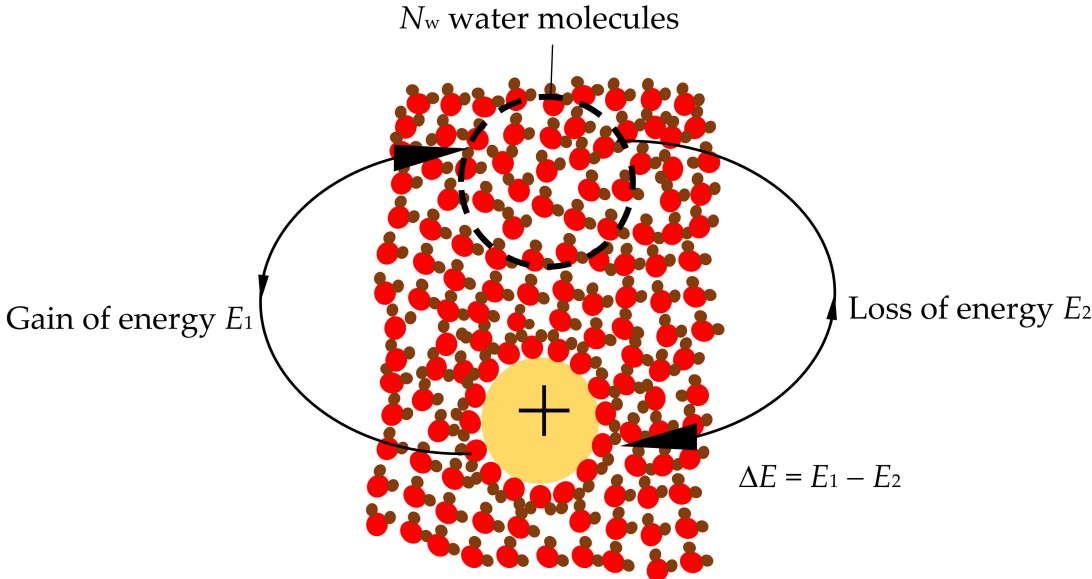

**Figure 2.** Mechanism of ion adsorption on the air/water interface and its concurrent energetical balance.

To make our approach clearer, we processed Equations (11) and (12), thus obtaining separate expressions for the energies $E_1$ and $E_2$, which are shown in Figure 2. Hence, for kosmotropic ions we obtained the following relations:

$$E_1 = u_i^s - u_i^b = -\frac{7}{6}\frac{\pi \rho_w L_{iw}}{R_{ih}^3} + \frac{4}{3}\frac{\pi \rho_w L_{iw}}{R_{ih}^3} \tag{19}$$

$$E_2 = u_w^s - u_w^b = -\frac{7}{6}\frac{\pi \rho_w L_{ww}}{R_{ih}^3} + \frac{4}{3}\frac{\pi \rho_w L_{ww}}{R_{ih}^3} \tag{20}$$

For chaotropic ions, we obtained the following relations:

$$E_1 = u_i^s - u_i^b = -\frac{2}{3}\frac{\pi \rho_w L_{iw}}{R_{ih}^3}\left(1 + \frac{3}{4}\frac{R_i}{R_{ih}}\right) + \frac{4}{3}\frac{\pi \rho_w L_{iw}}{R_{ih}^3} \tag{21}$$

$$E_2 = u_w^s - u_w^b = -\frac{2}{3}\frac{\pi \rho_w L_{ww}}{R_{ih}^3}\left(1 + \frac{3}{4}\frac{R_{ib}}{R_{ih}}\right) + \frac{4}{3}\frac{\pi \rho_w L_{ww}}{R_{ih}^3} \tag{22}$$

We wish everyone could be able to repeat and reproduce our calculations. For this reason, we provide the whole information needed for these calculations. The data are presented in CGS system, which we used in our calculation. We present our final output data in SI system.

Table 2 presents the parameters of the water, which we used in our theoretical study.

**Table 2.** Polarizability, ionic potential, and radius of the water molecule and density of the of the water molecules in the bulk of water [43,65,66].

| Parameters | $\alpha_w$, Å$^3$ | $I_w$, Erg | $R_w$, Å | $\rho_w$, cm$^{-3}$ |
|---|---|---|---|---|
| Water | 1.480 | $2.020 \times 10^{-11}$ | 1.380 | $3.330 \times 10^{22}$ |

The ionic parameters of $OH^-$, $HCO_3^-$, and $H_3O^+$ ions are presented in Table 3. One can see that the value of $N_w$ is not integer number and even can be smaller than one. This looks strange at first

glance, but it does not suffer of lack of logics because after the integration, the water is observed as a structureless continuum, whose surface energy is larger than its bulk energy.

**Table 3.** Polarizability, ionization potential, bare radius, hydration radius [65,66], ensemble of $N_w$ surface water molecules (see Equation (A11)) and type of ion for $OH^-$, $HCO_3^-$ and $H_3O^+$ ions.

| Ion | $OH^-$ | $HCO_3^-$ | $H_3O^+$ |
|---|---|---|---|
| $\alpha_0$, $Å^3$ | 2.040 | 4.320 | 1.210 |
| $I_i$, Erg | $3.450 \times 10^{-12}$ | $5.900 \times 10^{-12}$ | $2.110 \times 10^{-9}$ |
| $R_{ib}$, Å | 1.330 | 1.560 | 1.300 |
| $R_{ih}$, Å | 2.110 | 2.140 | 2.120 |
| $N_w$ | 0.895 | 1.444 | 0.835 |
| Type | kosmotrope | chaotrope | kosmotrope |

Table 4 presents the London constants $L_{iw}$ and $L_{ww}$, the dimensionless gain of energy of transfer of one ion from the bulk to the air/water interface $E_1/k_BT$, the dimensionless loss of energy of transfer of $N_w$ surface water molecule to the bulk $E_2/k_BT$, the dimensionless total energy of adsorption of one ion to the air/water interface $(E_1 - E_2)/k_BT$, and the adsorption energy $E_{Ads}$ of 1 mol ions on the air/water interface for $OH^-$, $HCO_3^-$ and $H_3O^+$ ions.

**Table 4.** London constants $L_{iw}$ and $L_{ww}$, dimensionless gain of energy of transfer of the ion from the bulk to the air/water surface $E_1/k_BT$, dimensionless loss of energy of transfer of $N_w$ water molecules from the air/water interface to the bulk of the water $E_2/k_BT$, dimensionless total energy of adsorption of one ion $(E_1 - E_2)/k_BT$ and adsorption energy $E_{Ads}$ per mol of ions on the air/water interface for $OH^-$, $HCO_3^-$ and $H_3O^+$ ions.

| Ion | $L_{iw}$, $cm^6$ | $L_{ww}$, $cm^6$ | $\frac{E_1}{k_BT}$ | $\frac{E_2}{k_BT}$ | $\frac{E_1-E_2}{k_BT}$ | $E_{Ads}$, kJ/mol |
|---|---|---|---|---|---|---|
| $OH^-$ | $1.33 \times 10^{-59}$ | $2.97 \times 10^{-59}$ | 0.61 | 1.36 | −0.75 | 1.83 |
| $HCO_3^-$ | $4.38 \times 10^{-59}$ | $4.79 \times 10^{-59}$ | 3.49 | 3.82 | −0.33 | 0.33 |
| $H_3O^+$ | $5.37 \times 10^{-59}$ | $2.77 \times 10^{-59}$ | 2.43 | 1.25 | 1.18 | −2.87 |

Shown in Figure 3 are the energies of adsorption of $OH^-$, $H_3O^+$, and $HCO_3^-$ ions on the air/water interface. One can see in Table 4 and Figure 3 that the transfer of ion from the bulk to the air/water interface always gains energy ($E_1/k_BT > 0$), i.e., the ion overcomes the image repulsion of the air/water interface. Moreover, one can see that the transfer of $N_w$ surface water molecules into the bulk always losses energy ($E_2/k_BT > 0$), i.e., these water molecules interact with significantly larger amount of other water molecules when they are located in the bulk, compared to their state on the air/water interface. In this way they decrease their energy. One can see in Table 4 as well that for the case of $OH^-$ and $HCO_3^-$ ions the loss of energy is larger than the gain of energy. Hence, the total change of energy is negative, i.e., the system decreases its energy when they adsorb on the air/water interface. Therefore, their adsorption on the air/water interface is energetically favorable. On the contrary, the gain of energy is larger than the loss of energy for the case of $H_3O^+$ ions. Hence, their adsorption on the air/water interface is energetically unfavorable. This means that there is an attraction force between the $OH^-$ and $HCO_3^-$ ions and the air/water interface, while the $H_3O^+$ ions exhibit repulsion force by the same surface. One can see as well the values of their adsorption energies per mol ions. These values are significantly smaller than the adsorption energies of the surfactants. For example, the adsorption energy of sodium dodesyl sulfate (SDS), which is common well known surfactant, is $E_{Ads} = 44.12$ kJ/mol [67]. The adsorption energies are heat effects of the adsorption of these ions. Therefore, these values represent the changes of the enthalpies $\Delta H$ of the ions during their adsorption, i.e., there is a negative change of the entalpy ($\Delta H < 0$) for case of the $OH^-$ and $HCO_3^-$ ions and positive change of the entalpy ($\Delta H > 0$) for the case of $H_3O^+$ ions. Nevertheless, the negative change of the entalpy ($\Delta H < 0$) is a required but not sufficient condition for spontaneous process. The positive change

of the entalpy ($\Delta H > 0$) is required but not sufficient condition for forced process. The change of the entropy is important to know as well.

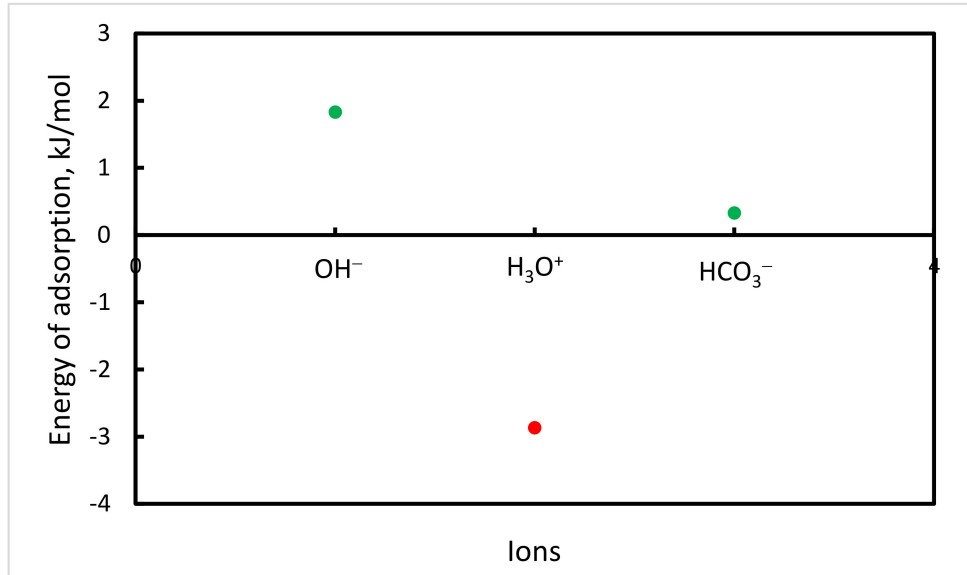

**Figure 3.** Energies of adsorption of $OH^-$, $H_3O^+$, and $HCO_3^-$ ions on the air/water interface.

### 3.3. Entropy of Adsorption of $OH^-$, $HCO_3^-$, and $H_3O^+$ Ions on the Air/Water Interface

The entropy of adsorption of ion from the bulk onto the air/water interface is calculated via Equations (15)–(17).

The change of the entropy of transfer of $N_w$ surface water molecules into the bulk is negligible as far as the water molecules do not move, but only vibrate due to the hydrogen bond network. For this reason, we only calculate the entropy of transfer of the ions from the bulk onto the air/water interface. Shown in Table 5 are the values of change of the entropy of adsorption of $OH^-$, $HCO_3^-$, and $H_3O^+$ ions and the parameters needed for their calculation. One can see that the ions decrease their entropy values if they adsorb on the air/water interface.

**Table 5.** Parameters needed for calculation of the change of entropy of the adsorption of $OH^-$, $HCO_3^-$, and $H3O^+$ ions on the air/water interface and the calculated values of the latter.

| Ion | $\Gamma$, m$^{-2}$ | $v$, m$^{-3}$ | $m$, kg | $\overline{c_{3D}^2}$, m$^2$/s$^2$ | $\overline{c_{2D}^2}$, m$^2$/s$^2$ | $S_{2D}$, J/K | $S_{3D}$, J/K | $\Delta S$, J/K |
|---|---|---|---|---|---|---|---|---|
| $OH^-$ | $6.023 \times 10^{13}$–$6.023 \times 10^{15}$ | $3.8 \times 10^{18}$ | $2.82 \times 10^{-26}$ | $2.03 \times 10^{-9}$ | $3.05 \times 10^{-9}$ | $2.82 \times 10^{-6}$ | $2.93 \times 10^{-3}$ | $-2.28 \times 10^{-3}$ |
| $HCO_3^-$ | $6.023 \times 10^{13}$–$6.023 \times 10^{15}$ | $9.52 \times 10^{20}$ | $1.01 \times 10^{-25}$ | $2.01 \times 10^{-9}$ | $3.01 \times 10^{-9}$ | $2.98 \times 10^{-6}$ | $0.71$ | $-0.71$ |
| $H_3O^+$ | $6.023 \times 10^{13}$–$6.023 \times 10^{15}$ | $9.58 \times 10^{20}$ | $3.15 \times 10^{-26}$ | $2.02 \times 10^{-9}$ | $3.04 \times 10^{-9}$ | $2.83 \times 10^{-6}$ | $0.69$ | $-0.69$ |

Once we have the enthalpy and entropy values of the adsorption of the $OH^-$, $HCO_3^-$, and $H_3O^+$ ions calculated, we can calculate the related changes of the free Gibbs energy by means of Equation (18).

Shown in Figure 4 is the change of the Gibs free energy during the adsorption of $OH^-$, $H_3O^+$, and $HCO_3^-$ ions on the air/water interface. Table 6 shows the values of the changes of the free Gibbs energy of adsorption of $OH^-$, $HCO_3^-$, and $H_3O^+$ ions on the air/water interface. One can see in both Figure 4 and Table 6 that the $OH^-$ and $HCO_3^-$ ions decrease their free Gibbs energy if they adsorb on the air/water interface. Hence, their adsorption is a spontaneous process. On the contrary, the $H_3O^+$ ions increase their free Gibbs energy if they adsorb on the air/water interface. They can adsorb on the air/water interface only if they are forced to adsorb. We must recognize that these calculations are valid for ions either in deionized (DI) water or in diluted salt solutions, in which the Ray–Jones effect [5,6,68,69] is in power (up to 0.002 mol/L). At larger salt concentrations, the adsorption of ions on the air/water interface becomes energetically unfavorable due to an increase of the image repulsion

force. The ions located in the bulk interact not only with the water molecules but also with the other ions located in the bulk as well. In DI water or diluted salt solution, the image repulsion is not so strong yet. This allows the adsorption of ions on the air/water interface. Therefore, if we follow the general thermodynamic logics and the values of $\Delta G$ in Table 6, we arrive at the conclusion that the $OH^-$, $HCO_3^-$ ions adsorb on the air/water interface, while the $H_3O^+$ ions are repelled by the latter. Thus, an electrical double layer (EDL) is formed at the very bubbles–negatively charged adsorption layer of $OH^-$, $HCO_3^-$ ions and positively charged diffuse layer of $H_3O^+$ ions.

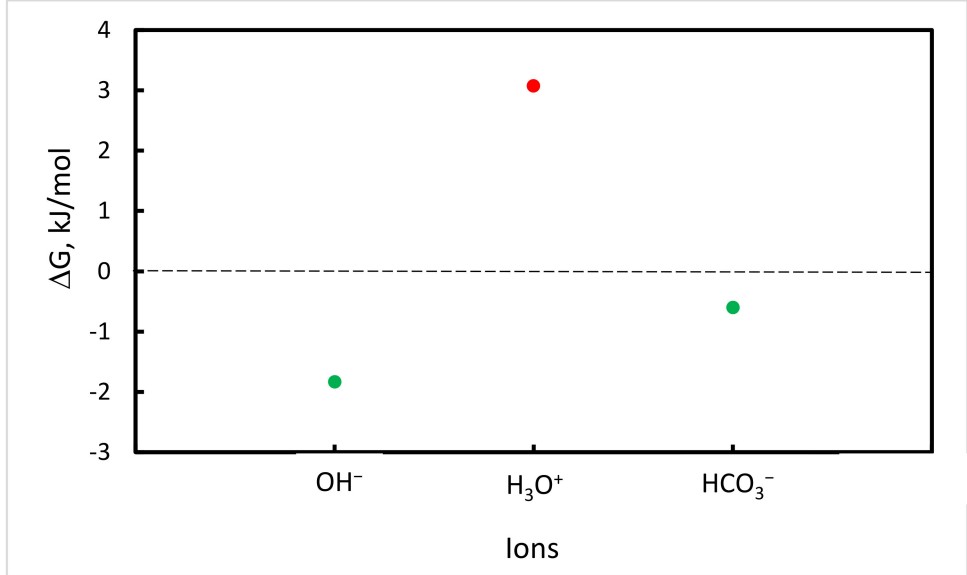

**Figure 4.** The change of the Gibbs free energy during the adsorption of $OH^-$, $H_3O^+$, and $HCO_3^-$ ions on the air/water interface.

**Table 6.** Values of change of the free Gibbs energy of adsorption of the $OH^-$, $HCO_3^-$, and $H_3O^+$ ions on the air/water interface.

| Ion | $\Delta G$, kJ/mol |
|---|---|
| $OH^-$ | −1.83 |
| $HCO_3^-$ | −0.60 |
| $H_3O^+$ | 3.07 |

We should note here that the zeta potential of gas bubbles in water, measured by the different authors, varies within some reasonable limits [10–21] depending on the method of measurement, but all of them agree on the negative value of the zeta potential of the micro-(or nano) bubbles in fresh DI water and its salt solution until the isoelectric point. Our theoretical analysis on the ion adsorption on the air/water interface accounts for the dispersion intermolecular forces and is valid for bubbles with every size. Therefore, the $OH^-$ and $HCO_3^-$ shall adsorb on the surface of the bubbles, while the $H_3O^+$ will be repelled by the bubble despite its size.

### 3.4. The Law of Parsimony and the Negative Charge of the Bubbles

The negative charge of the bubbles is a 100-year-old scientific problem. A number of explanations were suggested in the literature to solve it. They are complex, and some of them contradict to each other. How to arrive at the right explanation? We search the help of the "Occam's Razor" philosophical principle known as "The Law of Parsimony".

The basic principle was pronounced by the Greek philosopher Aristotle (384–322 BC) ("The more limited, if adequate, is always preferable") and by the Roman mathematician and astronomer

Claudius Ptolemy (100–170 AC) ("We consider it a good principle to explain the phenomena by the simplest hypothesis possible"). The English mathematician and physicist Isaac Newton (1643–1727) stated "We are to admit no more causes of natural things than such as are both true and sufficient to explain their appearances". Nevertheless, it is assumed that the roots of this principle in its present form lie in the works of the English Franciscan friar William of Occam (1287–1347) [70] who formulated "Never posit pluralities without necessity" ("Numquam ponenda est pluralitas sine necessitate"). The Franciscan scholastic theological philosopher John Punch (1603–1661) further formulated "Entities are not to be multiplied without necessity" ("Entia non sunt multiplicanda praeter necessitate"). The cottish metaphysical philosopher William Hamilton (1788–1856) called this maximum in 1852 as "Occam's razor" or "The Law of Parsimony". This principle was reconsidered in the light of the contemporary scientific knowledge and technology by the American philosopher Jonathan Schaffer in 2014 [71] who pointed out that the "razor" is too blunt measure for the ontological economy, failing to distinguish fundamental from derivative entities. Hence, he replaced the "razor" with a "laser", which is focused specifically on fundamental entities and commands: "Do not multiply fundamental entities without necessity!" Therefore, Schaffer had in mind that "Occam's razor" should be applied mostly for fundamental entities, having intrinsic nature. Hence, we formulated as a fundamental problem the negative charge of the bubbles in water. We scrutinize the experimental findings on this problem and the related different explanations in the search for the simples one. To be more specific, we present in Figure 5 the scheme of the experimental findings and their explanations in the literature and the present study as well.

As shown in Figure 5, the basic entity in form of scientific puzzle consists of experimental knowledge that (i) "the bubbles have negative zeta potential depending on the pH and the ionic strength values", (ii) "the nonionic surfactants practically do not affect the zeta potential while the ionic surfactants do", and (iii) "pairs of bubbles from $CO_2$ in contact do not coalesce for a long time on the contrary of pairs of bubbles from other gases". Evidently, these findings have one common origin causing negative charge of the bubbles. The question is "what makes them negatively charged?" The right explanation should be a fundamental entity (simple, sufficient, and definite explanation), which should be cause of our basic entity, consisted of its three above mentioned statements. In addition, the right explanation should not introduce new puzzles. Hence, our basic entity should be undoubtfully derivative from the fundamental entity (the right explanation). We will apply Occam's razor to the explanations from the literature and our own as well.

Reference [28] suggests a hypothesis that even the purest water contains anionic surface active traces. They developed theoretical model based on this hypothesis, explaining the Ray–Jones effect [6,69] and the negative charge of the bubbles. Nevertheless, their model contradicts the experimental fact that the nonionic surfactants do not affect the zeta potential of the bubbles. Hence, this explanation has two flaws: (i) the hypothesis needs a proof, and (ii) the theoretical model only partially explains our basic entity (statements (i), (ii), (iii)). Therefore, Occam's razor cuts this hypothesis and its related consequences.

Another explanation suggested by Reference [36] consists of charge transfer from the bulk toward the surface water molecules due to topologically defected hydrogen bond network. This explanation is supported by quantum-chemical calculations. Therefore, we accept that such phenomenon is possible. But this explanation opens the door to another puzzle—in which way the topological defects of the hydrogen bond network are related with the pH and the ionic strength values and the presence of carbon dioxide ($CO_2$) in water? If we accept this explanation, we accept a new puzzle. Hence, Occam's razor cuts the multiplication of the entities as not justified in this particular case.

Another well accepted explanation supported by References [22–24,72] suggests a hypothesis that the autolysis of the surface water is different from the autolysis of the bulk water. They developed a theory based on this hypothesis resulting in positive adsorption of both the $OH^-$ and $H_3O^+$ ions on the air/water interface. Afterwards, they suggested another hypothesis based on their theory that the adsorption of $OH^-$ ions is larger than the adsorption of $H_3O^+$ ions, thus resulting in total

negative charge of the bubbles. Overall, this explanation suggests two hypotheses, which need proofs. Occam's razor cuts these hypotheses.

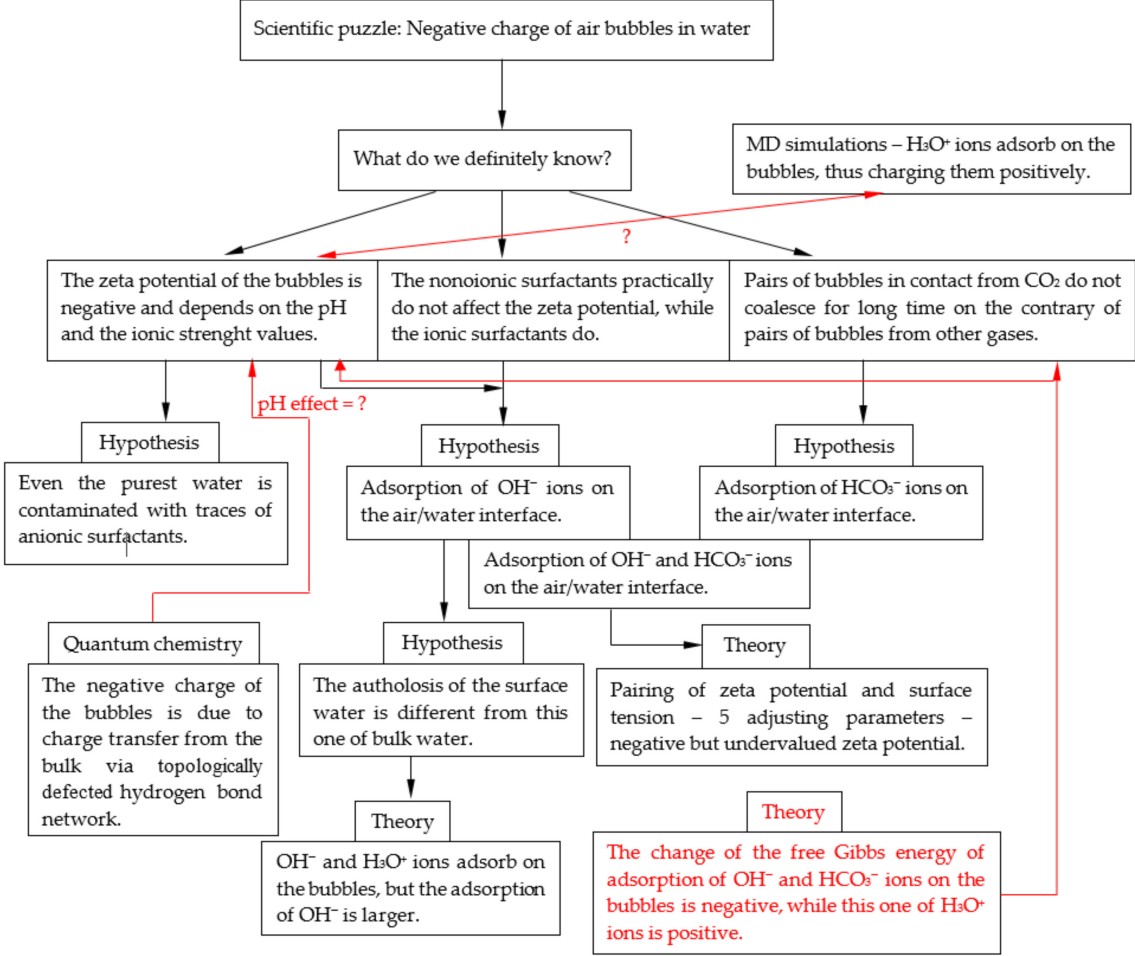

**Figure 5.** A scheme of the basic experimental findings and the available explanations of the origin of the negative charge of the bubbles.

Another explanation [26,27] suggested a theoretical model pairing the surface tension and the zeta potential, taking into account all the possible ions into the water with variable pH. The model operates with the energy of interaction of the ions with the air/water interface and the specific length for each ion, at which the concentration gradient of the ion is developed. These two parameters are to be adjusted by means of the experimental value of the zeta potential versus pH. Thus developed, the model results in positive adsorption of the ions charging the air/water interface despite being either $H_3O^+$ ions charging the bubbles positively at low pH or $OH^-$ ions charging the bubbles negatively at higher pH values. In addition, to calculate zeta potential values equal to the experimental ones, the model imposes high values of the energy of adsorption of the ions close to these ones of the weakest surfactants and huge subsurface concentration. Otherwise, the predicted zeta potential values are significantly lower than the experimental ones. Thus explained, the physical reason for the adsorption of the ions on the air/water interface remains unclear. In other words, this explanation supports the hypothesis that the $OH^-$ ions adsorb on the bubbles at moderate and high pH values, and the $H_3O^+$ ions adsorb on the bubbles at low pH values. Unfortunately, the adjustment of theoretical energy of interaction and specific ionic length to experimental data of the zeta potential is not a proof of the hypothesis, because the physical reason for this adsorption remains obscure. Occam's razor cuts this hypothesis.

Other explanations [9,37,38] based on completely experimental studies suggest the hypothesis that both $HCO_3^-$ and $OH^-$ ions adsorb on the air/water interface thus charging it negatively. When the pH value is increased, both the $HCO_3^-$ and $OH^-$ ions increase their concentrations, and this correlates with larger in absolute value negative zeta (or surface) potential. On the contrary, when the pH value is decreased, both the $HCO_3^-$ and $OH^-$ ions decrease their concentrations, and this correlates with smaller in absolute value negative zeta (or surface) potential, isolectric point (IEP) in the pH range of 3 to 4 and positive zeta potential at pH < 3. These experimental observations support the hypothesis of adsorption of the ions on the air/water interface, but unfortunately, they are not equal to a proof of the hypothesis. Occam's razor cuts this hypothesis.

The majority of molecular dynamic simulations (MDS) of the air/water interface [29–33] have shown that the $OH^-$ ions should not be adsorbed at the air/water interface, but $H_3O^+$ ions should, thus charging the bubbles positively. Few authors argued that their MDS analysis [34,35] shows the opposite. The results of the MD simulations contradict the experimental findings. Moreover, some of the MD simulations contradict other MD simulations. Occam's razor cuts these explanations due their contradiction to the experimental data.

We suggest in the present work a thermodynamic approach to solve the puzzle. It is well known that the spontaneous processes decrease the free Gibbs energy of a given system [48]. On the contrary, the forced processes increase the free Gibbs energy of the system. We calculate the change of the free Gibbs energy of adsorption of $HCO_3^-$, $OH^-$, and $H_3O^+$ ions and found out that the $HCO_3^-$ and $OH^-$ ions spontaneously adsorb, while the $H_3O^+$ ions adsorb only when being forced on the air/water interface. We set the limit of validity of the exploited theory up to ionic strength values of about 0.002 mol/L. A possible flaw of this explanation is that the theory always predicts a negative value of the zeta potential although being variable, while at pH < 3 the experiment shows positive values of the zeta potential. Under such conditions, the ionic strength has values of the of order 0.002 mol/L and larger, which violates the validity of the theory. Therefore, we set the validity of the theory in the pH range of 3 < pH < 11 and ionic strength up to 0.002 mol/L. Another possible flaw of this theory is that the energies of adsorption of the ions are close to the thermal energy $k_BT$, which should result in very low values of the zeta (or surface) potential if the latter is calculated by means of the classical Poisson–Boltzmann equation. From Occam's razor's viewpoint, we suggest a fundamental entity causing the basic entity with its three statements and with no hypotheses left. We state that the $HCO_3^-$ and $OH^-$ ions adsorb on the bubbles, while the $H_3O^+$ ions are repelled by them at ionic strength less than 0.002 mol/L. Keeping in mind the age and the toughness of the puzzle, we leave the door open for other fundamental explanations.

## 4. Conclusions

We applied the law of parsimony to shed light on an old scientific problem: the negative charge of the bubbles in water. The intensive experimental studies on this topic resulted in different explanations in the literature [9,20,22–24,26–33,36–38,72]. We scrutinized the basic literature in the light of the Occam's razor philosophical principle and showed clearly its flaws. Moreover, we suggest a new explanation by thermodynamic analysis, thus proving that the $HCO_3^-$ and $OH^-$ adsorb on the surface of the bubbles, while the $H_3O^+$ ions are repelled by them. Therefore, an electrical double layer (EDL) is formed with a negatively charged air/water interface and a positively charged diffuse layer consisting of $H_3O^+$ ions. The theory is valid for ionic strength lower than 0.002 mol/L or for the pH range of 3 < pH < 11. The adsorption energies of the ions are close to the thermal energy $k_BT$, which indicates just minor adsorption with very small values of the zeta potential if the latter is calculated by means of the classical Poisson–Boltzmann equation.

Finally, we open the door to other possible fundamental explanations of the negative charge of the air bubbles in water.

**Author Contributions:** Conceptualization, S.I.K. and N.A.G.; methodology, S.I.K.; software, N.A.G.; validation, S.I.K. and N.A.G.; formal analysis, S.I.K.; investigation, S.I.K. and N.A.G.; resources, N.A.G.; data curation, N.A.G.; writing—S.I.K.; writing—review and editing, S.I.K.; visualization, N.A.G. and S.I.K.; supervision, S.I.K.; project administration, S.I.K.; funding acquisition, S.I.K. All authors have read and agreed to the published version of the manuscript.

**Funding:** This paper is supported by European Union's Horizon 2020 research and innovation program under grant agreement No. 821265, project FineFuture (Innovative technologies and concepts for fine particle flotation: unlocking future fine-grained deposits and Critical Raw Materials resources for the EU.

**Conflicts of Interest:** The authors declare no conflict of interest.

## Nomenclature

Latin

$A = \sqrt{\alpha/\pi}$

| | |
|---|---|
| $A_v$ | Empirical constant determined by Marcus; |
| $C_{\mathrm{OH^-}}$ | Concentration of $\mathrm{OH^-}$ ions; |
| $C_{\mathrm{H_3O^+}}$ | Concentration of $\mathrm{H_3O^+}$ ions; |
| $C_{\mathrm{HCO_3^-}}$ | Concentration of $\mathrm{HCO_3^-}$ ions; |
| $C_{\mathrm{CO_3^{2-}}}$ | Concentration of $\mathrm{CO_3^{2-}}$ ions; |
| $\overline{c^2_{(2D)}}$ | Average square rate of one ion on the air/water interface; |
| $\overline{c^2_{(3D)}}$ | Average square rate of one ion on the bulk of water; |
| $c_i$ | Instant speed of particle "i" from the system; |
| $D$ | Bulk diffusion coefficient; |
| $D_s$ | Surface diffusion coefficient; |
| $E_1$ | Gain of energy during the adsorption of one ion on the air/water interface; |
| $E_2$ | Loss of energy during the desorption of $N_w$ water molecules from the air/water interface to the bulk of water; |
| $\Delta G$ | Change of free Gibbs energy of adsorption of the ions on the air/water interface; |
| $H$ | Ha function of Boltzmann; |
| $\Delta H$ | Change of enthalpy of adsorption of the ions on the air/water interface; |
| $I_i$ | Ionization potential of ion "i"; |
| $I_j$ | Ionization potential of ion "j"; |
| $I_w$ | Ionization potential of one water molecule; |
| $k_B$ | Boltzmann constant; |
| $K_{a1}$ | First dissociation constant of carbonic acid; |
| $K_{a2}$ | Second dissociation constant of carbonic acid; |
| $K_w$ | Ionic constant of water; |
| $K_r$ | Equilibrium constant of dissolving of $\mathrm{CO_2}$ in water; |
| $L_{iw}$ | London constant of dispersion interaction of one ion "i" with one water molecule; |
| $L_{ww}$ | London constant of dispersion interaction of one water molecule with ensemble of $N_w$ water molecules; |
| $L_{ij}$ | London constant of dispersion interaction between particles i and j; |
| $m$ | Mass of one ion; |
| $N$ | Total number of particles in the system; |
| $N_i$ | Number of particles in the phase cell "i"; |
| $N_w$ | Ensemble water molecules, suitable in a volume equal to the volume of one bulk ion; |
| $n_w$ | Hydration number of one ion; |
| $R_{ib}$ | Bare radius of ion "i"; |
| $R_{ih}$ | Hydration radius of ion "i"; |
| $R_w$ | Radius of one water molecule; |
| $r$ | Radius vector originating from the center of the ion toward the bulk; |
| $r_{ij}$ | Distance between particles i and j; |
| $S$ | Entropy; |
| $\Delta S$ | Change of entropy of adsorption of the ions on the air/water interface; |
| $S_{(2D)}$ | Entropy of the ions on the air/water interface; |

## Nomenclature

| | |
|---|---|
| $S_{(3D)}$ | Entropy of the ions on in the bulk of water; |
| $T$ | Absolute temperature; |
| $t$ | Time; |
| $u_{i0}$ | Specific adsorption energy of adsorption of ion "i" on the air/water interface; |
| $u_{iw}^s$ | Dispersion energy of interaction of surface ion "i" on the air/water interface with the whole bulk of water; |
| $u_{iw}^b$ | Dispersion energy of interaction of bulk ion "i" with the whole bulk of water; |
| $u_{ww}^s$ | Dispersion energy of interaction of ensemble of $N_w$ surface water molecules, suitable in a volume equal to the volume of one bulk ion, with the bulk of water underneath the interphase boundary; |
| $u_{ww}^b$ | Dispersion energy of interaction of ensemble of $N_w$ bulk water molecules, suitable in a volume equal to the volume of one bulk ion, with the whole bulk of water; |
| $\Delta u_i$ | $u_{iw}^s - u_{iw}^b$; |
| $\Delta u_w$ | $u_{ww}^b - u_{ww}^s$; |
| $W$ | Thermodynamic probability; |
| $\overline{x^2}$ | Average square displacement of diffusing particle; |
| $z$ | Vertical coordinate; |
| Greek | |
| $\alpha = m/2k_BT$ | |
| $\alpha_i$ | Polarizability of ion "i"; |
| $\alpha_w$ | Polarizability of one water molecule; |
| $\Gamma$ | Adsorption of the ions on the air/water interface; |
| $\lambda$ | Number of phase cells; |
| $\mu$ | Dynamic viscosity of coefficient of water; |
| $\nu$ | Concentration of the ions in the bulk; |
| $v_w$ | Volume of one water molecule; |
| $\rho_w$ | Density of water. |

## Appendix A.

*Appendix A.1. Theory of Ivanov for the Adsorption of Ions on the Air/Water Interface*

As mentioned above, the specific adsorption energy of the ions on the air/water interface is based on the exchange of ion from the bulk with the surface water molecules (see Equation (10)).

Furthermore, an important property of each ion located in water is its hydration. For monovalent ions, Marcus [65] found that the hydration number $n_w$ of the ions can be represented by the empirical relation:

$$n_w = \frac{A_v}{R_{ib}} \tag{A1}$$

where $A_v = 3.6$ Å for all ions, and $R_{ib}$ is the bare radius of the ion. He further assumed that the hydrating $n_w$ water molecules, considered as spheres with radius $R_w = 1.38$ Å and volume $v_w = 11$ Å$^3$, are squeezed around the ion, forming a layer of thickness $R_{ih} - R_{ib}$ and volume:

$$n_w v_w = \frac{4\pi}{3}\left(R_{ih}^3 - R_{ib}^3\right) \tag{A2}$$

where $R_{ih}$ is the radius of the hydrated ion. The last relation can be used to calculate $R_{ih}$.

A special accent [42,73] is placed on the compressibility of the hydration shells—there are certain ions, called kosmotropes [74,75] (or structure making, e.g., Li$^+$, Na$^+$, F$^-$), whose hydration shells do not deform upon their adsorption at the air/water interface, while the hydration shells of all the other ions, called chaotropes [74,75] (or structure breaking, e.g., K$^+$, Cl$^-$, Br$^-$, NO$_3^-$, etc.), redistribute in such a way during their adsorption at the air/water interface that the upper part (toward the air) of the ion becomes bare, while the lower part (toward the bulk) becomes over-occupied with hydrated water molecules (see Figure A1).

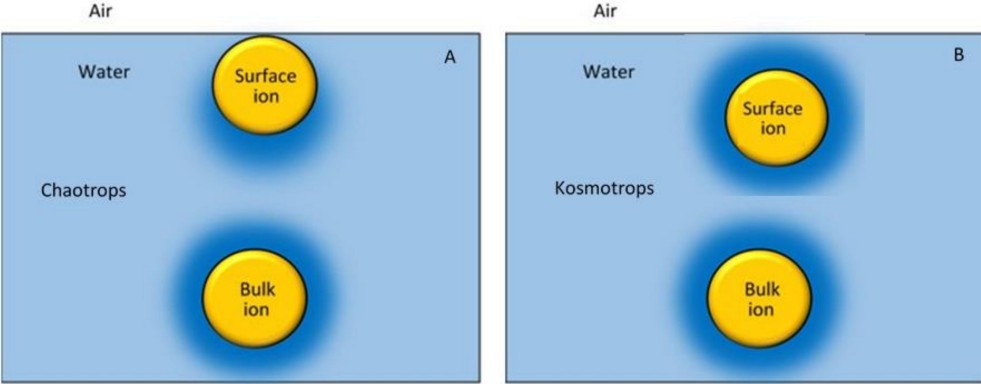

**Figure A1.** Examples of chaotrops (**A**) and kosmotrops (**B**)—the ions that are chaotrops deform their hyddration shells during their adsoprtion, while the ions that are kosmotrops keep their hydration shell undeformed during their adsoprtion.

As mentioned above, the theory operates with the dispersion interaction, which is the most ubiquitous interaction in the world. The dispersion interaction between species *i* and *j* at a distance $r_{ij}$ [76] can be presented by the following relation:

$$u_{ij} = -\frac{L_{ij}}{r_{ij}^6} \tag{A3}$$

where the London constant $L_{ij}$ is related to the static polarizabilities $\alpha_i$ and $\alpha_j$ and the ionization potentials $I_i$ and $I_j$ of the interacting species:

$$L_{ij} = \frac{3\alpha_i\alpha_j}{2}\frac{I_iI_j}{I_i+I_j} \tag{A4}$$

We first calculate the energy of interaction $u_{iw}^S$, depicted in Equation (10), for the chaotropic type of ions situated at the air/water interface with the whole bulk of water (see Figure A2). For this purpose, $u_{ij}$ from Equation A3 is integrated over the volume of the water excluding the hydration shell, with $r_{ij}$ being the distance between the volume element $dr$ and the ion positioned at $r = 0, z = 0$ (that is, the integration is over $z > -R_{ib}$ and $r^2 + z^2 < R_{ih}^2$).

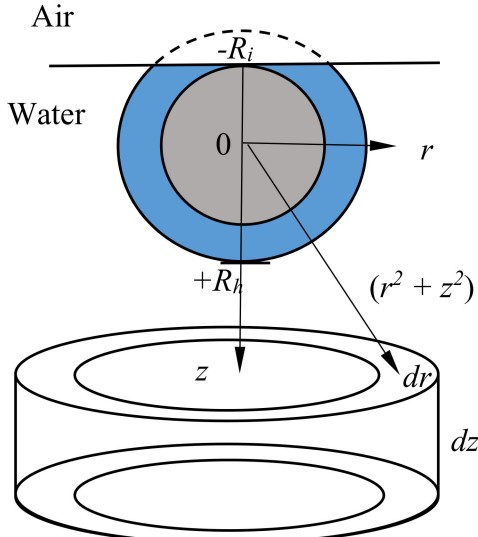

**Figure A2.** Illustration of the integration procedure applied to derive the energy of interaction of surface ion with the whole bulk of water.

The integration is performed in cylindrical coordinates:

$$u_{iw}^{S} = -\int_{-R_{ib}}^{R_{ih}} \int_{\sqrt{R_{ih}^2 - z^2}}^{\infty} \frac{L_{iw}\rho_w 2\pi r \mathrm{d}r \mathrm{d}z}{(r^2 + z^2)^3} - \int_{R_{ih}}^{\infty} \int_{0}^{\infty} \frac{L_{iw}\rho_w 2\pi r \mathrm{d}r \mathrm{d}z}{(r^2 + z^2)^3} = -\frac{2\pi}{3}\frac{L_{iw}\rho_w}{R_h^3}\left(1 + \frac{3}{4}\frac{R_{ib}}{R_{ih}}\right) \tag{A5}$$

where $\rho_w$ is the particle density of water. Similarly, the energy $u_i^B$ of the interaction of bulk ion with the whole bulk of water (integration in spherical coordinates) is given by:

$$u_{iw}^{B} = -\int_{R_{ih}}^{\infty} \frac{L_{iw}}{r_{iw}^6}\rho_w 4\pi r_{iw}^2 \mathrm{d}r_{iw} = -\frac{4\pi}{3}\frac{L_{iw}\rho_w}{R_{ih}^3} \tag{A6}$$

The respective energies $u_{ww}^S$ and $u_{ww}^B$ of the ensemble of water molecules (assumed to be in a sphere of radius $R_{ih}$, or a part of it) are

$$u_{ww}^{S} = -\frac{2\pi}{3}\frac{L_{ww}\rho_w}{R_{ih}^3}\left(1 + \frac{3}{4}\frac{R_{ib}}{R_{ih}}\right); u_{ww}^{B} = -\frac{4\pi}{3}\frac{L_{ww}\rho_w}{R_{ih}^3} \tag{A7}$$

Substituting Equations (A5)–(A7) into Equation (10) (see the main text) for $u_{i0}$, one obtains an explicit relation of the adsorption energy of the chaotropic ions (called here ions of type I) on the air/water interface:

$$u_{i0} = \left(1 - \frac{3}{4}\frac{R_{ib}}{R_{ih}}\right)\frac{2\pi}{3}\frac{\rho_w}{R_{ih}^3}(L_{iw} - L_{ww}) \tag{A8}$$

To calculate $u_{i0}$ for the kosmotropic ions (called here ions of type II), one must set $R_{ih} = R_{ib}$ in Equation (A8), which simplifies the expression to

$$u_{i0} = \frac{\pi}{6}\frac{\rho_w}{R_{ih}^3}(L_{iw} - L_{ww}) \tag{A9}$$

The London constants $L_{iw}$, for the interaction ion-water molecule, and $L_{ww}$, for the interaction of $N_w$ water molecules with a single water molecule, are calculated directly from Equation (A4):

$$L_{iw} = \frac{3\alpha_i\alpha_w}{2}\frac{I_i I_w}{I_i + I_w}; L_{ww} = \frac{3}{4}N_w\alpha_w^2 I_w \tag{A10}$$

For the calculation of $L_{ww}$, the ensemble of $N_w$ water molecules is regarded as a sphere with polarizability $N_w\alpha_w$ [42]. The number $N_w$ is assumed to be equal to the ratio between the volume of the bare ion and the volume of one water molecule [57]:

$$N_w = \frac{R_i^3}{R_w^3} \tag{A11}$$

where $R_w = 1.38$ Å corresponding to the volume of the water molecule 11 Å$^3$ [42]. The value used for the static polarizability of water was $\alpha_w = 1.48$Å$^3$ and for the ionization potential was $I_w = 2.02 \times 10^{-18}$ J [66]. For the cations, we used the second ionization potential, since the first one corresponds to the ionization of the respective atoms, not ions. Since the anions have already accepted one extra electron, their ionization potential must be equal to the negative value of the electron affinity.

We calculated the adsorption energies of OH$^-$, HCO$_3^-$, and H$_3$O$^+$ ions on the air/water interface using Equations (A8) and (A9) as far as HCO$_3^-$ is chaotropic, while OH$^-$ and H$_3$O$^+$ are kosmotropic ions. We took the values of the ionic radii, polarizabilities and ion potentials from References [66] and [77]. The adsorption energies are heat effects of adsorption of the above ions on the air/water interface indicating wheatear the adsorption of certain ion is either exothermic or endothermic. These heat effect are the enthalpy changes $\Delta H$, which will be used to calculate the Gibbs free energies of the adsorption of the above mentioned ions on the air/water interface.

*Appendix A.2. Entropy Change of Adsorption of Ions on the Air/Water Interface*

We will derive firstly an expression about entropy of the ions in the bulk, and after this, we will continue with the ions on the surface. The thermodynamic probability $W$ is defined by:

$$W = \frac{N!}{\prod\limits_{i=1}^{\lambda} N_i!} \tag{A12}$$

where $N$ is the total number of the particles in the system, $N_i$ is the number of particles in a phase cell "$i$" and $\lambda$ is number of the phase cells. The thermodynamic probability counts for the number of the different ordering of $N$ particles distributed in $\lambda$ phase cells. Hence, following the Sterling approximation $\ln x! = x\ln x - x$ ($x$ is very large number) after some algebraic transformation we obtain:

$$\ln W = N \ln N - \sum_{i=1}^{\lambda} N_i \ln N_i \tag{A13}$$

We have $N$ particles distributed in $\lambda$ phase energetical cells in accord with the universal Maxwell-Boltzmann distribution:

$$N_i = \nu A^3 \exp\left[-\frac{mc_i^2}{k_B T}\right] \tag{A14}$$

where $N_i$ is the number of particles, located in phase cell "$i$", in which they have velocity $c_i$, $\nu$ is the number of particles per unit volume, $m$ is the mass of one particle, $A = \sqrt{\alpha/\pi}$, $\alpha = m/2k_B T$. At $\lambda \to \infty$ we have continuous (not discrete) variation of the velocity of the particles:

$$N = \nu f = \nu A^3 \exp\left[-\frac{mc^2}{k_B T}\right] \tag{A15}$$

where $f$ is the density of the probability distribution function. Therefore, we obtain from Equation (A13):

$$\ln W = -\iiint \nu f \ln \nu f \, du\, dv\, dw \tag{A16}$$

where $u, v, w$ are the velocities in the three directions of the space in Cartesian coordinate system. The first term of the right-hand site of Equation (A13) is neglected because the second term is significantly larger at $\lambda \to \infty$.

The Ha function of Boltzmann $H$ is defined by the expression:

$$\ln W = -H \tag{A17}$$

Hence, Equation (A16) acquires the form:

$$H = \iiint \nu f \ln \nu f \, du\, dv\, dw \tag{A18}$$

Furthermore, by substituting Equation (A15) into Equation (A18), we obtain

$$H = \iiint \nu f \left(\ln \nu A^3 - \alpha c^2\right) du\, dv\, dw \tag{A19}$$

Furthermore we process Equation (A19) via angebraic transformtions:

$$H = \nu \ln \nu A^3 \underbrace{\iiint f \, du\, dv\, dw}_{1} - \nu \alpha \underbrace{\iiint f c^2 \, du\, dv\, dw}_{\overline{c^2}} = \nu\left(\ln \nu A^3 - \alpha \overline{c^2}\right) \tag{A20}$$

where $\overline{c^2}$ is the average square speed of the particles. Equation (A20) can be presented in the following form:

$$H = \nu\left[\ln\left(\nu\left(\frac{m}{2k_B T\pi}\right)^{\frac{3}{2}}\right) - \frac{m}{2k_B T}\overline{c^2}\right] \tag{A21}$$

Furthermore, by processing Equation (A21), we obtain the following expressions for the density of the entropy in 3D and 2D states:

$$S_{(3D)} = \frac{3}{2} \nu k_B \left[ \ln \nu + \ln\left( \frac{m}{2k_B T \pi} \right) - \frac{m \overline{c^2_{(3D)}}}{3k_B T} \right] \tag{A22}$$

$$S_{(2D)} = \frac{3}{2} \Gamma k_B \left[ \ln \Gamma + \ln\left( \frac{m}{2k_B T \pi} \right) - \frac{m \overline{c^2_{(2D)}}}{3k_B T} \right] \tag{A23}$$

where $\Gamma$ is the number of the particles per unit area, $\overline{c^2_{(3D)}}$ and $\overline{c^2_{(2D)}}$ are average square $3D$ and $2D$ speeds of the particles. As far as Equations (A22) and (A23) regard the density of the entropy, Equation (A22) is formally multiplied by one cubic meter, while Equation (A23) is multiplied by one square meter, thus giving in both cases the entropy in the same dimension (*J/K*).

We used Equation (A22) to calculate the entropy of the ions in the bulk and Equation (A23) to calculate the entropy of the ions on the air/water interface. We took the values of $\nu$ for the different ions from the speciation analysis conducted in Reference [44]. To calculate the value of $S_{(2D)}$, one needs to know the adsorption $\Gamma$ of each ion. We do not know it, but we could roughly estimate its value, because if such an adsorption layer exists, it should be rarely populated. The value of $\Gamma$ for surfactant adsorption layer usually varies between $\Gamma = 10^{-8}$ mol/L and $\Gamma = 10^{-5}$ mol/L. For this reason, we assumed reasonable to accept the range of $\Gamma = 10^{-10} - 10^{-8}$ mol/L for adsorption values of $OH^-$, $HCO_3^-$, and $H_3O^+$ ions. Hence, we calculated the change of the entropy of the transfer of ions with concentration $\nu$ from the bulk to the air/water interface until they achieve adsorption value $\Gamma = 10^{-8}$ mol/L. Furthermore, we need information about the value of $\overline{c^2_{(3D)}}$ and $\overline{c^2_{(2D)}}$. First we will calculate the value of the bulk diffusion coefficient $D$ via the well-known Stokes formula valid spherical particles [41]:

$$D = \frac{k_B T}{6\pi\mu R_h} \tag{A24}$$

where $\mu$ is dynamic viscosity of water and $R_h$ is the hydration radius of the ion. The value of the surface diffusion coefficient $D_S$ is 1.5 times larger than this one of the bulk diffusion coefficient ($D_S = 1.5\,D$) [49]. Hence, we can calculate the average square rate of the ions via the Einstein relation [41]:

$$\overline{x^2} = 2Dt \tag{A25}$$

where $\overline{x^2}$ is the average square displacement of diffusing particle and $t$ is time. Hence, from Equation (A25) one can calculate the average square speed of the particle $\overline{c^2}$:

$$\overline{c^2} = \frac{\overline{x^2}}{t} = 2D \tag{A26}$$

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
