# Peer review of "The Law of Parsimony and the Negative Charge of the Bubbles"

_coatings, doi:10.3390/coatings10101003_

Round 1

Reviewer 1 Report

The manuscript entitled The law of parsimony and the negative charge of the bubbles is theoretical study of air-water interface. I commented as follows;
1.Section 2 Theoretical review is very long. This is the key section of this study, but the author should present it in a more summary manner.
2.The symbols for multiplication should be properly indicated.
3.The results should be presented as a graph, not a table.
4.As for the usual papers, many references are cited. However, most of them are not mentioned in detail. The author should only cite references that are specific.
5.It is difficult to understand what the difference is between the surface potential characteristics of fine bubbles and those of others. It should be mentioned whether the author considers them to be different characteristics or the same. In the first place, we should mention the size of the bubble we are interested in.
6.The many symbols and letters are used. The author should summarize them as nomenclatures.

Author Response

Dear Reviewer 1,

We would like to thank you for your assessment of our manuscript and for your critical comments.

We believe that the new version of our manuscript and the response to your notes will satisfy you.

Please find here bellow detailed response to your comments.

With best regards,

Stoyan Karakashev

The manuscript entitled The law of parsimony and the negative charge of the bubbles is theoretical study of air-water interface. I commented as follows;

I commented as follows;
1.Section 2 Theoretical review is very long. This is the key section of this study, but the author should present it in a more summary manner.

We agree. We shortened the theoretical part of the manuscript, but made an appendix where the reader can follow the detailed development of the theory. We did this for completeness reasons.

2.The symbols for multiplication should be properly indicated.

To our knowledge, the symbols for multiplication in the manuscript are in the common standard accepted in the scientific literature. We may have misunderstood this comment, for which we are sorry.

3.The results should be presented as a graph, not a table.

We added two graphs with results.

4.As for the usual papers, many references are cited. However, most of them are not mentioned in detail. The author should only cite references that are specific.

We carefully scrutinized the cited literature. The cited papers are grouped in the following groups: (i) number of papers, which agree on a certain experimental fact; (ii) papers suggesting explanation on scientific puzzle; (iii) pioneering papers; The reviewer must agree that only groups (ii) and (iii) are worthy for analysis and discussion, while group (i) shows the intensification of the research in a given area.

5.It is difficult to understand what the difference is between the surface potential characteristics of fine bubbles and those of others. It should be mentioned whether the author considers them to be different characteristics or the same. In the first place, we should mention the size of the bubble we are interested in.

Our theoretical analysis treats the physico-chemical properties of the air/water interface. The equations derived are valid for air/water interface and the its curvature does not matter. i.e. the theory is valid for bubbles with every size.

6.The many symbols and letters are used. The author should summarize them as nomenclatures.

We agree. We added list of symbols (nomenclature).

Reviewer 2 Report

The paper deals with the origin of negative electrical charge of an air/water interface. The fact that the air/water interface in negatively charged is relatively well proved experimentally, however, no comprehensive and profound explanation on the mechanisms of this phenomenon is given in the literature. The Authors decided to deal with this problem scientifically and philosophically (Occam’s razor), combining these two approaches to obtain reasonable conclusion. They analyzed carefully the interactions between ions and water molecules in the bulk and at the interface. To estimate the adsorption propensity of charge-inducing ions they used the theory of Ivanov to calculate the change in the Gibbs free energy and determined for each ion, if the adsorption can be spontaneous or not. The philosophical approach helped the Authors to discuss the validity of other possible hypotheses of the charge origin and identify that this one proposed by them in the paper (derived on the basis of well explained and logically presented calculations) is the most probable.

            The article is very well written and presents many very useful and interesting conclusions. In my opinion it is ready for publication, after addressing some minor points, listed below:

1. The H3O+ and the OH- ions attract each other. Simultaneously, the OH- ions are attracted by the water/air interface while the H3O+ ions are repelled (as was shown in the paper). Is it possible to compare the forces of interactions between both ions and between ions and interface to show directly the magnitude of forces and proof that the forces of repulsion for interface/H3O+ are larger than attraction between OH-/H3O+?

2. Line 433 – The Authors should reconsider the possibility of presence of anionic surfactant in water. As was shown recently (Vakarelski et al. Langmuir 2018, 34:2096; Krasowska et al. J. Phys. Chem. C 2019, 123:3645), the clean system (perfectly pure water) practically does not exist. In my opinion the Occam’s razor cuts the hypothesis given in this section, because there is no way to differentiate between the surfactant type (non-ionic or anionic). The Authors should consider commenting on this issue.

Typos:

3. line 62 – “that” is repeated twice

4. line 120 – in the section title “ios”, should be “ions”

5. line 329 – “the” is repeated twice

6. line 428 – typo: “chgarged”, it should be charged

Author Response

Dear Reviewer 2,

We would like to thank you for your assessment of our manuscript and for your comments.

We believe that the new version of our manuscript and the response to your notes will satisfy you.

Please find here bellow detailed response to your comments.

With best regards,

Stoyan Karakashev

The paper deals with the origin of negative electrical charge of an air/water interface. The fact that the air/water interface in negatively charged is relatively well proved experimentally, however, no comprehensive and profound explanation on the mechanisms of this phenomenon is given in the literature. The Authors decided to deal with this problem scientifically and philosophically (Occam’s razor), combining these two approaches to obtain reasonable conclusion. They analyzed carefully the interactions between ions and water molecules in the bulk and at the interface. To estimate the adsorption propensity of charge-inducing ions they used the theory of Ivanov to calculate the change in the Gibbs free energy and determined for each ion, if the adsorption can be spontaneous or not. The philosophical approach helped the Authors to discuss the validity of other possible hypotheses of the charge origin and identify that this one proposed by them in the paper (derived on the basis of well explained and logically presented calculations) is the most probable.

1. The H3O+and the OH- ions attract each other. Simultaneously, the OH- ions are attracted by the water/air interface while the H3O+ ions are repelled (as was shown in the paper). Is it possible to compare the forces of interactions between both ions and between ions and interface to show directly the magnitude of forces and proof that the forces of repulsion for interface/H3O+ are larger than attraction between OH-/H3O+?

This is an interesting and worthy to investigate problem. The reviewer means the electrostatic interaction, which is governed by different power law of the dispersion interaction. Our rough calculation shows that the average distance between the OH- ions on the air/water interface is in order of 13 nm (G»10-8 mol/m2). This corresponds to energy of repulsion between them about 130 J/mol.  The Debye length is about 1/k = 240 nm. Therefore the electrostatic energy of attraction between OH- and H3O+ is about 7 J/mol. These electrostatic energies of interaction we compare with the dispersion energies of attraction of OH- ions and repulsion of H3O+ to the air/water interface. We can assume them equal to their adsorption energies, which are 1.83 kJ/mol for OH- ions and 2.87 kJ/mol for the H3O+ ions. Concussively we have 130 J/mol energy of repulsion between OH- ions on the air/water interface versus 1.83 kJ/mol energy of attraction of  OH- ions with the air/water interface; we have about 7 J/mol energy of attraction between the OH- (on the air/water interface) and H3O+ ions (in the diffuse layer) versus 2.87 kJ/mol energy of repulsion of the H3O+ from the air/water interface.

2. Line 433 – The Authors should reconsider the possibility of presence of anionic surfactant in water. As was shown recently (Vakarelski et al. Langmuir 2018, 34:2096; Krasowska et al. J. Phys. Chem. C 2019, 123:3645), the clean system (perfectly pure water) practically does not exist. In my opinion the Occam’s razor cuts the hypothesis given in this section, because there is no way to differentiate between the surfactant type (non-ionic or anionic). The Authors should consider commenting on this issue.

The purest ordinary DI water is solution of carbonic acid containing CO2, HCO3- and CO32- ions along with the intrinsic ions of the water. The reviewer is familiar with the complex picture of the ion interactions with the air/water interface along with the specific orientation of the surface water molecules at the air/water interface forming surface capacitor. In accord with the Occam’s razor philosophy we should not multiply the entities without necessity. To define such a necessity, we should strive solve the task with the current entities. I am not sure that the task is completely solved or completely branded as unsolvable. The future will show if such a necessity will be defined. Otherwise we don’t state that there is no any other contaminations even in the purest possible water, but a solid proof is needed.

Typos:

3. line 62 – “that” is repeated twice

4. line 120 – in the section title “ios”, should be “ions”

5. line 329 – “the” is repeated twice

6. line 428 – typo: “chgarged”, it should be charged

Thank you. We made the correction.

Round 2

Reviewer 1 Report

The revisions are satisfied as reviewers' comments.